# Replication Study: Melanoma exosomes educate bone marrow progenitor cells toward a pro-metastatic phenotype through MET

**Jeewon Kim[1], Amirali Afshari[2†], Ranjita Sengupta[2†], Vittorio Sebastiano[1,3,4], Archana Gupta[2], Young H Kim[2], Reproducibility Project: Cancer Biology***

[1]Stanford Transgenic, Knockout and Tumor Model Center, Stanford Cancer Institute, California, United States; [2]System Biosciences LLC, California, United States; [3]The Institute for Stem Cell Biology and Regenerative Medicine, Stanford, United States; [4]Department of Obstetrics and Gynecology, Stanford School of Medicine, Stanford, United States

***For correspondence:**
tim@cos.io;
nicole@scienceexchange.com

[†]These authors contributed equally to this work

**Group author details:**
Reproducibility Project: Cancer Biology See page 14

**Abstract** As part of the Reproducibility Project: Cancer Biology we published a Registered Report (Lesnik et al., 2016) that described how we intended to replicate selected experiments from the paper 'Melanoma exosomes educate bone marrow progenitor cells toward a pro-metastatic phenotype through MET' (Peinado et al., 2012). Here we report the results. We regenerated tumor cells stably expressing a short hairpin to reduce Met expression (shMet) using the same highly metastatic mouse melanoma cell line (B16-F10) as the original study, which efficiently downregulated Met in B16F10 cells similar to the original study (Supplementary Figure 5A; Peinado et al., 2012). Exosomes from control cells expressed Met, which was reduced in exosomes from shMet cells; however, we were unable to reliably detect phosphorylated Met in exosomes. We tested the effect of exosome-dependent Met signaling on primary tumor growth and metastasis. Similar to the results in the original study, we did not find a statistically significant change in primary tumor growth. Measuring lung and femur metastases, we found a small increase in metastatic burden with exosomes from control cells that was diminished when Met expression was reduced; however, while the effects were in the same direction as the original study (Figure 4E; Peinado et al., 2012), they were not statistically significant. Differences between the original study and this replication attempt, such as level of knockdown efficiency, cell line genetic drift, sample sizes, study endpoints, and variability of observed metastatic burden, are factors that might have influenced the outcomes. Finally, we report meta-analyses for each result.
DOI: https://doi.org/10.7554/eLife.39944.001

## Introduction

The Reproducibility Project: Cancer Biology (RP:CB) is a collaboration between the Center for Open Science and Science Exchange that seeks to address concerns about reproducibility in scientific research by conducting replications of selected experiments from a number of high-profile papers in the field of cancer biology (Errington et al., 2014). For each of these papers a Registered Report detailing the proposed experimental designs and protocols for the replications was peer reviewed and published prior to data collection. The present paper is a Replication Study that reports the results of the replication experiments detailed in the Registered Report (Lesnik et al., 2016) for a paper by Peinado et al., and uses a number of approaches to compare the outcomes of the original experiments and the replications.

In 2012, Peinado et al. reported that exosomes isolated from highly metastatic murine melanoma cells (B16-F10) increased the metastatic burden of the primary tumors compared to exosomes from poorly metastatic melanomas (B16-F1) or control liposomes. Hepatocyte growth factor receptor (Met) was identified as a highly expressed protein in B16-F10 exosomes. Reduction of Met, and phosphorylated Met (pMet) by shRNA in B16-F10 exosomes resulted in a statistically significant decrease in lung and bone metastatic burden compared to controls (*Peinado et al., 2012*). Exosomes derived from melanoma cells were proposed to promote metastasis by education of bone marrow-derived cells through horizontal transfer of exosomal Met in order to prime the pre-metastatic niche and increase vascularization (*Peinado et al., 2012*).

The Registered Report for the paper by Peinado et al. described the experiments to be replicated (Figure 4E and Supplementary Figures 1C and 5A), and summarized the current evidence for these findings (*Lesnik et al., 2016*). Since that publication, additional studies have reported in different models that tumor derived exosomes administered to mice prior to tumor cell injection increased metastatic burden by inducing pre-metastatic niche formation (*Costa-Silva et al., 2015*; *Fong et al., 2015*; *Hoshino et al., 2015*; *Liu et al., 2016*; *Plebanek et al., 2017*; *Zhou et al., 2014*). Additionally, using the same tumor model as *Peinado et al., 2012*, Met expression was found to be heterogeneous in B16-F10 cells, with higher lung metastatic burden, and lower primary tumor burden, in cells expressing high levels of Met compared to cells expressing low levels of Met (*Adachi et al., 2016*). Injection of exosomes from high Met expressing cells increased the metastatic burden of low Met expressing cells (*Adachi et al., 2016*). The molecules present on tumor-derived exosomes, such as the specific repertoire of integrins, are important in dictating metastatic organotropism (*Hoshino et al., 2015*). There have been numerous studies that have aimed to identify exosomal cargo (e.g. proteins) (*Keerthikumar et al., 2016*). Similar to the study by *Peinado et al., 2012*, other studies, using various techniques (e.g. proteomic profiling, reverse phase protein array), have identified MET in exosomes from melanoma cells (*Lazar et al., 2015*; *Steenbeek et al., 2018*) as well as in hepatocellular carcinoma cells (*He et al., 2015*), neuroblastoma cells (*Keerthikumar et al., 2015*), ovarian cancer cells (*Liang et al., 2013*), sera from healthy donors and prostate cancer patients (*Cannistraci et al., 2017*), 293 T cells (*Li et al., 2016*), and as a fusion protein (PTPRZ1-MET) in glioblastoma cells (*Zeng et al., 2017*), while other studies did not identify MET in exosomes from breast cancer patients (*Chen et al., 2017*) nor in some melanoma cells (*Lazar et al., 2015*).

The outcome measures reported in this Replication Study will be aggregated with those from the other Replication Studies to create a dataset that will be examined to provide evidence about reproducibility of cancer biology research, and to identify factors that influence reproducibility more generally.

## Results and discussion

### Generation and characterization of shMet B16-F10 cells and exosomes

To test the effect exosome-dependent Met signaling has on primary tumor growth and metastasis, we used the same highly metastatic mouse melanoma cell line (B16-F10) and the same lentiviral system as the original study to make B16-F10 cells stably expressing an shRNA targeting *Met* (shMet) or a control shRNA (shScr) using the same targeting sequences as the original study. The experimental approach to generate and characterize the stable cells and isolated exosomes was described in Protocol 1 and 2 of the Registered Report (*Lesnik et al., 2016*). We tested various multiplicity of infection (MOI) ratios, all of which displayed expression of the shRNA with corresponding decreased *Met* and Met levels in shMet cells compared to shScr cells (*Figure 1—figure supplement 1*). We planned to utilize cells generated with an MOI of 10, similar to the original study, but observed that the Met levels in the shScr cells at this MOI were, for unknown reasons, decreased when compared to the shScr cells generated at the other MOI ratios (*Figure 1—figure supplement 1C*). Thus, we proceeded with the stable cells generated with an MOI of 20, which had 22.6% Met expression, and 25.1% phosphorylated Met (pMet) expression in the shMet cells relative to shScr cells (*Figure 1A–*

*C*). The stable cell lines generated in the original study were reported to have 64.1% Met expression and 23.4% pMet expression in the shMet cells relative to shScr cells (*Peinado et al., 2012*).

Using the same ultracentrifugation approach as the original study, we isolated exosomes from shScr and shMet cells. We confirmed the presence of the known exosome markers Hsc70, Tsg101, and Cd63 (*Figure 1D*) and found exosome number and size distribution quantified by NanoSight analyses to be similar between the two groups (*Table 1*). To test if Met and pMet were expressed in B16-F10 exosomes we performed Western blots and found that compared to shScr exosomes the

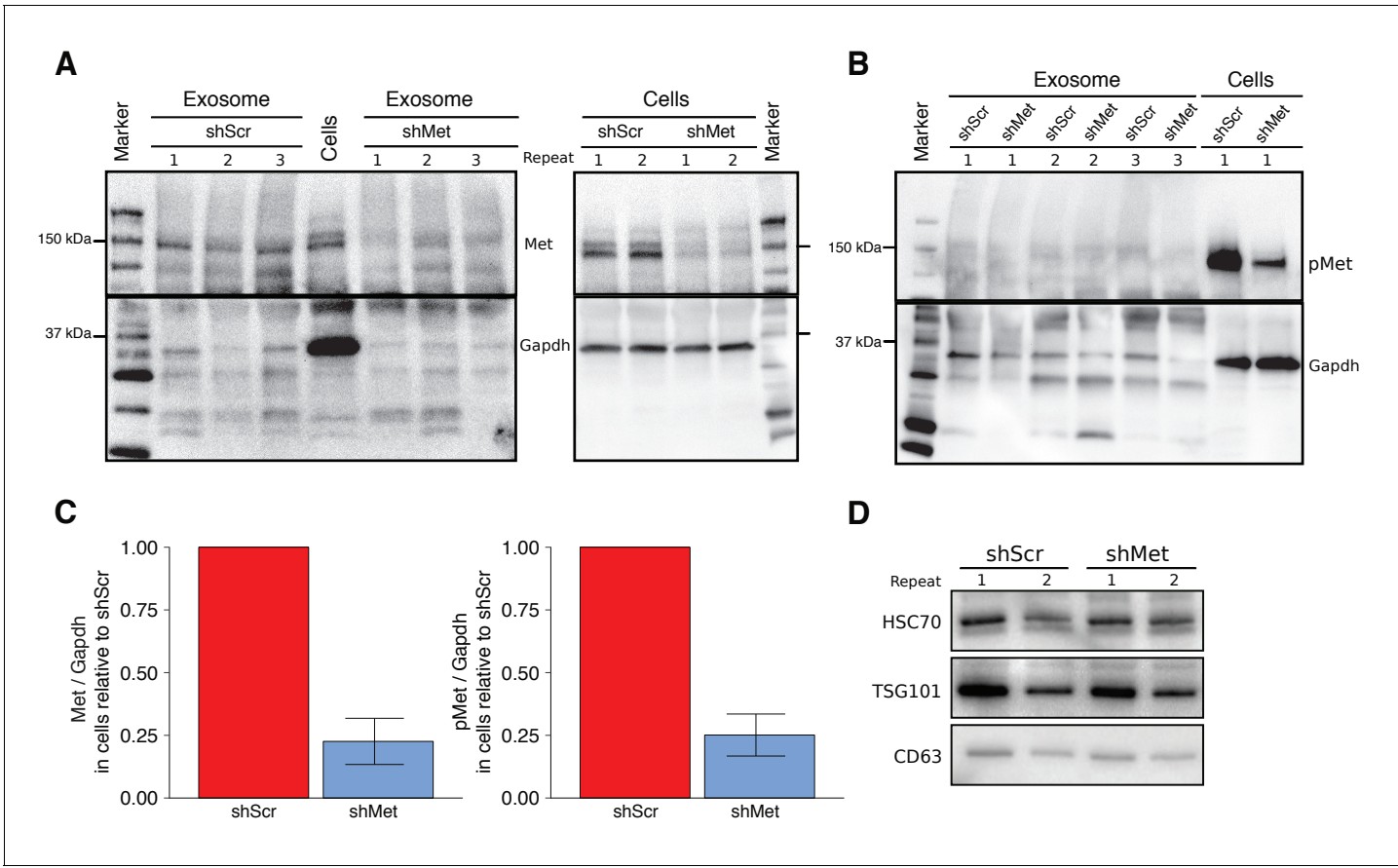

**Figure 1.** Characterization of shMet B16-F10 cells and exosomes. B16-F10 cells engineered to express shScr or shMet were used to purify exosomes. (A) Representative Western blots of exosomes and B16-F10 cells expressing the indicated shRNA were probed with antibodies specific for total Met (top panel) and Gapdh (bottom panel). Membranes were cut at ~75 kDa so that Met and Gapdh could be probed in parallel. Repeat indicates the number of independently isolated exosome and cell lysate preparations from the same batch of infected cells. The fourth lane, labeled 'Cells' are lysate from B16-F10 cells expressing shScr. (B) Representative Western blots of exosomes and B16-F10 cells expressing the indicated shRNA were probed with antibodies specific for phosphorylated (Tyr 1234/1235) Met (pMet) (top panel) and Gapdh (bottom panel). Membranes were cut at ~75 kDa so that pMet and Gapdh could be probed in parallel. Repeat indicates the number of independently isolated exosome preparations from the same batch of infected cells. (C) Western blot bands were quantified for cells. Met or pMet levels were normalized to Gapdh, and protein expression presented relative to shScr conditions. Expression level of shScr condition was assigned a value of 1. Means reported and error bars represent *SD.* Results are from 3 independent biological repeats for Met expression and 4 independent biological repeats for pMet expression. Exploratory analysis: one-sample *t*-test on Met levels (Met/Gapdh) in shMet cells compared to a constant of 1 (shScr cells): $t(2) = 8.41$, $p = 0.014$, Bonferroni corrected $p = 0.028$, Cohen's $d = 4.85$, 95% CI [0.55, 9.43]; one-sample *t*-test on pMet levels (Met/Gapdh) in shMet cells compared to a constant of 1 (shScr cells): $t(3) = 8.94$, $p = 0.003$, Bonferroni corrected $p = 0.006$, Cohen's $d = 4.47$, 95% CI [1.02, 8.01]. (D) Representative Western blot of exosomes isolated from cells expressing the indicated shRNA probed with exosome markers Hsc70, Tsg101, and Cd63 specific antibodies. Experiment performed on 3 independent biological repeats for each condition from the same batch of infected cells. Additional details for this experiment can be found at https://osf.io/aqm2m/.
DOI: https://doi.org/10.7554/eLife.39944.002

The following figure supplement is available for figure 1:

**Figure supplement 1.** Multiplicity of infection (MOI) ratios tested for stable cell line generation.
DOI: https://doi.org/10.7554/eLife.39944.003

**Table 1.** Nanosight analysis of exosomes.
Summary of exosome number and size distribution, with or without the finite track length adjustment (FTLA) algorithm, quantified by NanoSight analysis. All values (mean, *SD*, median, span, concentration (particles/ml)) are given as averages for preps generated during this study (n = 15 per cell line).

| | | Mean | Median | Sd | Span | Particles/mL |
|---|---|---|---|---|---|---|
| shMet | FTLA size distribution | 91.07 | 72.9 | 62.89 | 1.253 | 6.253e+11 |
| | size distribution | 91 | 75 | 65.44 | 1.334 | 6.253e+11 |
| shScr | FTLA size distribution | 88.47 | 71.8 | 56.04 | 1.211 | 9.477e+11 |
| | size distribution | 88.67 | 73.67 | 61.8 | 1.321 | 9.477e+11 |

DOI: https://doi.org/10.7554/eLife.39944.004

amount of Met expressed in shMet exosomes was decreased (*Figure 1A*); however, we were unable to reliably detect pMet expression in exosomes (*Figure 1B*). Importantly, this was not due to the inability of the antibody to detect pMet by Western blot, since pMet was detected in cell lysates run on the same gel. Additionally, as a preventative measure to block any residual phosphatase activity, we added inhibitors when preparing the isolated exosomes for electrophoresis. However, if there were carryover phosphatases present during isolation of exosomes by ultracentrifugation the pMet levels could have been diminished before the inhibitors were added. Furthermore, the exosomes were prepared for electrophoresis after first being stored at −20℃, instead of immediately after isolation, which might be critical for the detection of phospho-protein detection in exosomes. Inclusion of untransduced B16-F10 controls could have also indicated if unintended changes occurred in the stable cell lines utilized, especially given the potential impact puromycin selection can have on transcriptome profiles (*Guo et al., 2017*), and should be considered in the experimental design of future studies. The original study reported Met and pMet expression by Western blot in B16-F10 exosomes; however, the level of knockdown achieved in shMet exosomes was not reported. To summarize, we observed reduced Met and pMet expression in shMet cells compared to shScr cells, consistent with the stable cells reported in the original study; however, while we detected Met in exosomes from shScr cells, that was reduced in exosomes from shMet cells, we were unable to reliably detect pMet expression in isolated exosomes.

## Exosome-dependent Met signaling on primary tumor growth and metastasis

We next used shScr and shMet exosomes to replicate an experiment that tested whether exosome-dependent Met signaling impacted primary tumor growth and metastasis. Synthetic unilamellar 100 nm liposomes were used as a control to test if tumor exosomes enhance metastasis. This experiment is similar to what was reported in Figure 4E of *Peinado et al., 2012* and described in Protocol 3 in the Registered Report (*Lesnik et al., 2016*). Freshly isolated exosomes from shScr or shMet cells, or synthetic liposomes, were injected into C57BL/6 female mice three times a week for a total of four weeks, thereafter we implanted B16-F10 tumor cells engineered to express luciferase (B16-F10-luc). The planned study design involved waiting 21 days after B16-F10-luc tumor cell implantation before sacrificing the mice for analysis; however, two animals were found dead before this time point was reached (17 days after implantation), and of the surviving mice, the largest tumors reached >1 cm$^3$ prompting us to stop the experiment early (18 days after implantation). We measured primary tumor growth during the length of the study and observed increased growth among all the conditions (*Figure 2A*, *Figure 2—figure supplement 1A–D*). Interestingly, the primary tumors in mice injected with shScr exosomes were on average smaller than the primary tumors in mice injected with shMet or synthetic liposomes. This was also observed when measuring the weight of the dissected primary tumors at the end of the study (*Figure 2B*). There are a number of factors that can affect tumor growth, such as availability of nutrients, oxygen, and space that influence initial and continued growth of the tumor (*Cornelis et al., 2013*; *Talkington and Durrett, 2015*). There was not a statistically significant difference between the tumor weights of the three groups (Kruskal-Wallis: H(2) = 2.85, p=0.24). Additionally, we conducted two planned comparisons (shScr vs shMet; shScr vs synthetic liposomes), which were not statistically significant (see *Figure 2* figure legend). This is

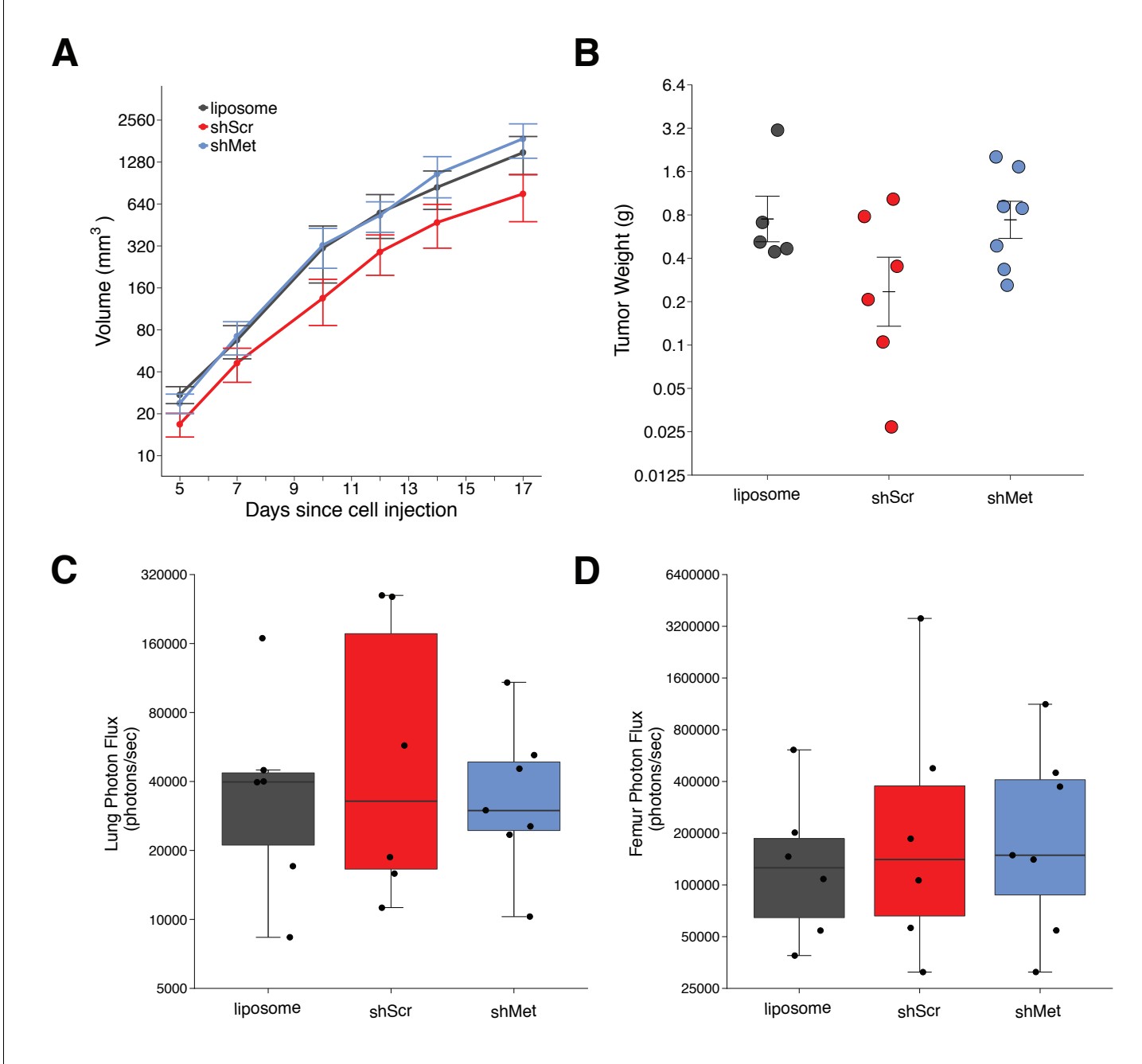

**Figure 2.** Primary tumor growth and metastatic burden of mice injected with exosomes. Female C57BL/6 mice were subcutaneously injected with B16-F10-luciferase cells after 4 weeks of intravenous injections (3 times a week) of shScr exosomes, shMet exosomes, or synthetic liposomes. (**A**) Following primary tumor detection caliper measurements were taken 3 times a week and used to calculate tumor volume. Line graph of tumor volume (y-axis is natural log scale) with means reported and error bars representing s.e.m. Number of mice monitored per group: synthetic liposomes = 7, shScr = 7, shMet = 7. Of note, not all mice had detectable primary tumors at the first measurement, one of the mice injected with synthetic liposomes never formed a detectable primary tumor despite the presence of metastatic burden, and two mice (one injected with B16-F19 shScr exosomes and one injected with synthetic liposomes) were found dead before the end of the study (day 17). Individual mouse tumor volume data is reported in *Figure 2 - figure supplement 1*. (**B**) At the end of the experiment (day 18), primary tumors were excised and weighed. Dot plot with means reported as crossbars and error bars represent s.e.m. Number of primary tumor weights per group: synthetic liposomes = 5, shScr = 6, shMet = 7. Kruskal-Wallis test on all three groups: H(2) = 2.85, $p$ = 0.24. Planned Wilcoxon-Mann-Whitney comparison between shScr and synthetic liposomes: $U$ = 22, $p$ = 0.247, Cliff's $d$ = 0.47, 95% CI [-0.28, 0.86]. Planned Wilcoxon-Mann-Whitney comparison between shScr and shMet: $U$ = 32, $p$ = 0.138, Cliff's $d$ = 0.52, 95% CI [-0.10, 0.85]. (**C**) Metastatic burden in lungs quantified by luciferin photon flux at 18 days after B16-F10-luc tumor cell injections. Box and whisker plot (y-axis is natural log scale) with median represented as the line through the box and whiskers representing values within 1.5 IQR of the first and third quartile.
*Figure 2 continued on next page*

*Figure 2 continued*

Individual data points represented as dots. Number of mice per group: synthetic liposomes = 6, shScr = 6, shMet = 7. One-way ANOVA on the luciferin photon flux values (natural log-transformed); $F_{(2,16)}$ = 0.226, uncorrected $p$ = 0.800 with *a priori* alpha level of 0.025, Bonferroni corrected $p > 0.99$. Planned contrast between shScr and synthetic liposomes; Fisher's LSD test; $t_{(16)}$ = 0.543, $p$ = 0.594 with *a priori* alpha level of 0.025, Bonferroni corrected $p > 0.99$, Cohen's $d$ = 0.31, 95% CI [-0.83, 1.45]. Planned contrast between shScr and shMet; Fisher's LSD test; $t_{(16)}$ = 0.620, $p$ = 0.544 with *a priori* alpha level of 0.025, Bonferroni corrected $p > 0.99$, Cohen's $d$ = 0.34, 95% CI [-0.76, 1.44]. (D) Metastatic burden in femurs quantified by luciferin photon flux at 18 days after B16-F10-luc tumor cell injections. Box and whisker plot (y-axis is natural log scale) with median represented as the line through the box and whiskers representing values within 1.5 IQR of the first and third quartile. Individual data points represented as dots. Number of mice per group: synthetic liposomes = 6, shScr = 6, shMet = 7. One-way ANOVA on the luciferin photon flux values (natural log-transformed); $F_{(2,16)}$ = 0.190, uncorrected $p$ = 0.829 with *a priori* alpha level of 0.025, Bonferroni corrected $p > 0.99$. Planned contrast between shScr and synthetic liposomes; Fisher's LSD test; $t_{(16)}$ = 0.573, $p$ = 0.575 with *a priori* alpha level of 0.025, Bonferroni corrected $p > 0.99$, Cohen's $d$ = 0.33, 95% CI [-0.82, 1.46]. Planned contrast between shScr and shMet; Fisher's LSD test; $t_{(16)}$ = 0.105, $p$ = 0.918 with *a priori* alpha level of 0.025, Bonferroni corrected $p > 0.99$, Cohen's $d$ = 0.06, 95% CI [-1.03, 1.15]. Additional details for this experiment can be found at https://osf.io/mzywk/.

DOI: https://doi.org/10.7554/eLife.39944.005

The following figure supplement is available for figure 2:

**Figure supplement 1.** Alternative visualizations of tumor growth and metastatic burden.

DOI: https://doi.org/10.7554/eLife.39944.006

consistent with the original study that stated no differences in primary tumor growth between the three groups; however, those data were not shown, preventing direct comparison of the original study results and the results from this replication attempt.

Metastatic burden was quantified by luciferin photon flux in the lungs and femurs of the mice from each group at the end of the study. Mice injected with the shScr exosomes achieved an average of $1.0 \times 10^5$ photons/sec in the lungs, which was reduced to $4.2 \times 10^4$ photons/sec in mice injected with shMet exosomes and $5.3 \times 10^4$ photons/sec in mice injected with synthetic liposomes (*Figure 2—figure supplement 1E*). Thus, the group means of the natural log transformed lung metastatic burden data were 10.8, 10.4, and 10.5 log units, respectively (*Figure 2C*). On average, there was a 36.9% natural log based percentage difference in lung metastatic burden between mice injected with shScr exosomes and shMet exosomes, and a 33.6% difference between mice injected with shScr and synthetic liposomes. To test if shMet exosomes or synthetic liposomes had decreased metastatic burden compared to shScr exosomes, we performed a one-way ANOVA on the luciferin photon flux values (natural log-transformed), which was not statistically significant ($F_{(2,16)}$ = 0.226, uncorrected $p$=0.800, Bonferroni corrected $p$>0.99). Additionally, we conducted two planned comparisons (shScr vs synthetic liposomes; shScr vs shMet), which were not statistically significant (see *Figure 2* figure legend). The original study reported metastatic burden in the lungs of mice injected with shScr exosomes [M = $5.1 \times 10^4$ photons/sec; 10.8 log units] were reduced by 165.3% [M = $1.3 \times 10^4$ photons/sec; 9.1 log units] and 80.0% [M = $2.3 \times 10^4$ photons/sec; 10.0 log units] with shMet exosomes and synthetic liposomes, respectively (*Peinado et al., 2012*). The range of metastatic lung values reported in the original study had a relative standard deviation (RSD) that were smaller (shScr = 2.9%; shMet = 9.9%; synthetic liposome = 3.8%) than the RSD observed in this replication attempt (shScr = 13%; shMet = 7.1%; synthetic liposome = 9.7%), particularly among the shScr and synthetic liposome groups.

For femurs, the average metastatic burden of mice injected with the shScr exosomes achieved an average of $7.4 \times 10^5$ photons/sec (12.2 log units), which was reduced to $3.3 \times 10^5$ photons/sec (12.1 log units) in mice injected with shMet exosomes and $1.9 \times 10^5$ photons/sec (11.7 log units) in mice injected with synthetic liposomes (*Figure 2D*, *Figure 2—figure supplement 1F*). So, on average, there was a 7.8% natural log based percentage difference in femur metastatic burden between mice injected with shScr exosomes and shMet exosomes, and a 44.3% difference between mice injected with shScr and synthetic liposomes, which was not statistically significant (see *Figure 2* figure legend). While the original study reported metastatic burden in the femurs of mice injected with shScr exosomes [M = $4.5 \times 10^4$ photons/sec; 10.4 log units], there was no reported metastatic burden in the femurs of mice injected with shMet exosomes or synthetic liposomes (*Peinado et al., 2012*). Additionally, similar to the lung values, the RSD for the metastatic femur values were smaller in the original study (shScr = 8.4%; shMet = 0%; synthetic liposome = 0%) than this replication attempt (shScr = 14%; shMet = 10%; synthetic liposome = 8.4%).

The higher RSD observed in this study is one of the factors that could influence if statistical significance is reached, particularly since the sample size of this replication attempt was determined *a priori* to detect the effect based on the originally reported data. Importantly, these results should take into consideration the experimental endpoint of this replication attempt, which was shorter than the original study and what was indicated in the Registered Report (18 days instead of 21 days), which could account for less macro-metastasis. Also, as mentioned above, inclusion of untransduced B16-F10 controls could have indicated if unintended changes to the cargo of exosomes from the stable cell lines utilized occurred and should be considered in the experimental design of future studies. To summarize, for this experiment we found results that were in the same direction as the original study and not statistically significant.

## Meta-analysis of original and replication effects

We performed a meta-analysis using a random-effects model, where possible, to combine each of the effects described above as pre-specified in the confirmatory analysis plan (*Lesnik et al., 2016*). To provide a standardized measure of the effect, a common effect size was calculated for each effect from the original and replication studies. Cliff's delta (*d*) is a non-parametric estimate of effect size that measures how often a value in one group is larger than the values from another group. It is used in this case because of violations to the assumptions of normality and equal variance in the original and replication studies. Importantly, the confidence interval around Cliff's *d* is asymmetric, while the *p* value is calculated using the normal distribution and is thus not well defined; however, there is no agreement on how to compute *p* values from an asymmetric distribution (*Dunne et al., 1996*; *Rohatgi and Saleh, 2000*). The estimate of the effect size of one study, as well as the associated uncertainty (i.e. confidence interval), compared to the effect size of the other study provides another approach to compare the original and replication results (*Errington et al., 2014*; *Valentine et al., 2011*). Importantly, the width of the confidence interval for each study is a reflection of not only the confidence level (e.g. 95%), but also variability of the sample (e.g. *SD*) and sample size.

There were a total of four comparisons made with the metastatic burden data from lungs and femurs of mice injected with shScr exosomes, shMet exosomes, or synthetic liposomes (*Figure 3*). For all comparisons, the results of the original study and this replication were consistent when considering the direction of the effect; however, the point estimate of the replication effect size was not within the confidence interval of the original result, and vice versa. The meta-analyses were not statistically significant for any of the effects (see *Figure 3* figure legend). Furthermore, the large confidence intervals of the meta-analyses along with statistically significant Cochran's *Q* tests (lung: shScr and liposomes, *p*=0.045; shScr and shMet, *p*=0.020; femur: shScr and liposomes, *p*=0.0084; shScr and shMet, *p*=0.0046) suggest heterogeneity between the original and replication studies.

This direct replication provides an opportunity to understand the present evidence of these effects. Any known differences, including reagents and protocol differences, were identified prior to conducting the experimental work and described in the Registered Report (*Lesnik et al., 2016*). However, this is limited to what was obtainable from the original paper and through communication with the original authors, which means there might be particular features of the original experimental protocol that could be critical, but unidentified. So while some aspects, such as cell line, mouse strain, and technique for exosome isolation were maintained, others were changed during the execution of the replication that could affect results, such as the time from cell injection after exosome education until euthanasia, which was shorter in this replication attempt than what was conducted in the original study (18 days instead of 21 days). Additionally, others were unknown or not easily controlled for. These include variables such as cell-intrinsic changes in Met expression and the associated gene expression profiles (*Adachi et al., 2016*), cell line genetic drift (*Ben-David et al., 2018*; *Hughes et al., 2007*; *Kleensang et al., 2016*), non-random genetic/transcriptional drift in the heterogeneous stable cells (*Guo et al., 2017*; *Shearer and Saunders, 2015*), genetic heterogeneity of mouse inbred strains (*Casellas, 2011*), the microbiome of recipient mice (*Macpherson and McCoy, 2015*), housing temperature in mouse facilities (*Kokolus et al., 2013*), and lot variability and quality of key reagents, such as the lentiviral particles (*Leek et al., 2010*). Environmental differences such as husbandry staff, bedding type and source, light levels, and other intangibles, all of which, by necessity, differed between the studies also affect experimental outcomes with mice (*Howard, 2002*; *Jensen and Ritskes-Hoitinga, 2007*; *Nevalainen, 2014*; *Sorge et al., 2014*). The difference in

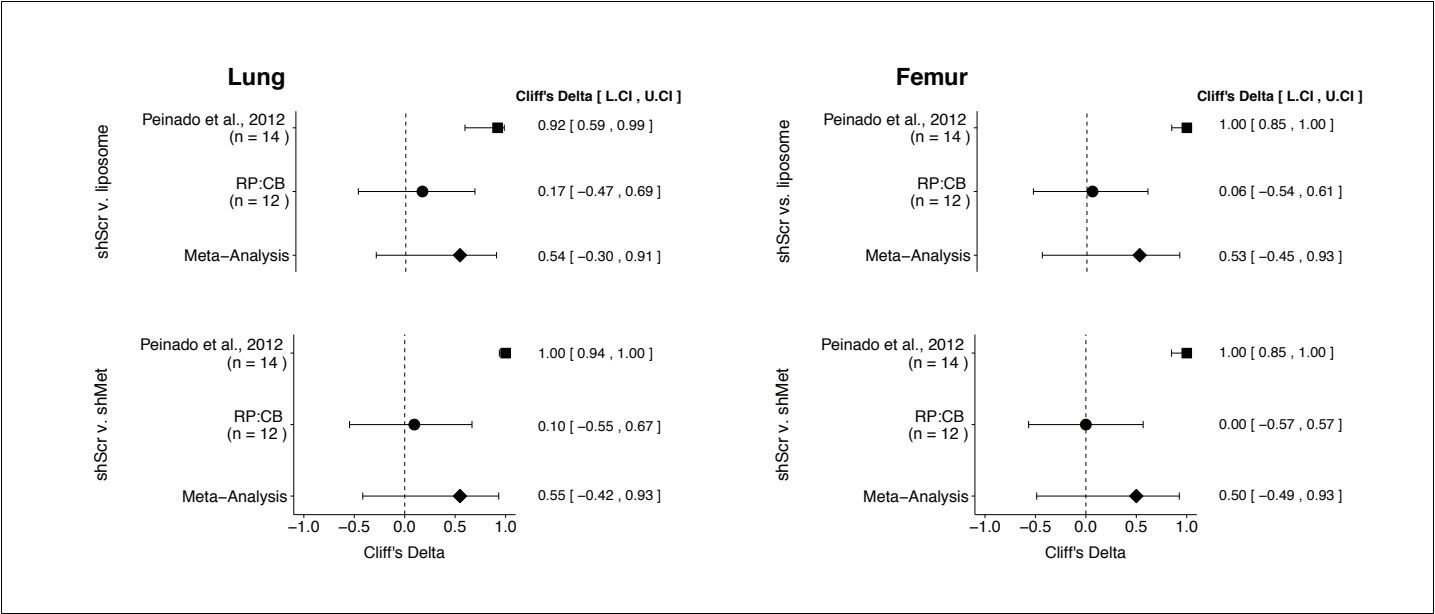

**Figure 3.** Meta-analyses of each effect. Effect size (Cliff's delta) and 95% confidence interval are presented for *Peinado et al., 2012*, this replication attempt (RP:CB), and a meta-analysis to combine the two effects. Cliff's delta is a standardized measure of how often a value in one group is larger than the values from another group, with a larger positive value indicating a decrease in metastatic burden in animals injected with either shMet exosomes or liposomes compared to animals injected with shScr exosomes. Random effects meta-analysis: lung metastatic burden (shScr v. liposome, *p* = 0.149; shScr v. shMet, *p* = 0.226) and femur metastatic burden (shScr v. liposome, *p* = 0.264; shScr v. shMet, *p* = 0.317). Sample sizes used in *Peinado et al., 2012* and this replication attempt are reported under the study name. Additional details for these meta-analyses can be found at https://osf.io/c69jx/. DOI: https://doi.org/10.7554/eLife.39944.007

sample sizes between the studies is also a factor that could limit the opportunity to detect a statistically significant difference, especially considering the higher variability observed in this replication attempt, as described above. The difference in achieved knockdown of Met in exosomes between the original study and this replication attempt is another important factor to consider. A higher level of knockdown might be required to yield a given phenotype with this experimental design. Importantly, observing and reporting all outcomes are informative to establish the range of conditions under which a given phenotype can be observed (*Bailoo et al., 2014*). Whether these or other factors influence the outcomes of this study is open to hypothesizing and further investigation, which is facilitated by direct replications and transparent reporting.

# Materials and methods

## Key resources table

| Reagent type (species) or resource | Designation | Source or reference | Identifiers | Additional information |
|---|---|---|---|---|
| Strain, strain background (*Mus musculus*, C57BL/6, female) | C57BL/6 | Charles River | Strain code: 027; RRID: IMSR_CRL:27 | |
| Cell line (*M. musculus*) | B16-F10 | doi:10.1038/nm.2753 | RRID: CVCL_0159 | |
| Cell line (*M. musculus*) | B16-F10-luc | doi:10.1038/nm.2753 | | expresses firefly luciferase |

*Continued on next page*

*Continued*

| Reagent type (species) or resource | Designation | Source or reference | Identifiers | Additional information |
|---|---|---|---|---|
| Other | synthetic uniloamellar 100 nm liposomes | Encapsula NanoSciences | | 13 mg/ml L-a-Phosplatydilcholine, 2.78 mg/ml cholesterol (7:3 molar ratio P:C) |
| Sequence-based reagent | pGIPZ non-silencing shRNA lentiviral control particles | Dharmacon/ GE Life Sciences | cat# RHS4348 | sense sequence: 5'-ATCTCGCTTGGG CGAGAGTAAG-3' |
| Sequence-based reagent | pGIPZ mouse *Met* shRNA lentiviral particles | Dharmacon/ GE Life Sciences | cat# VGM5520-200377256; clone ID: V3LMM_456078 | sense sequence: 5'-CCAGACTTTTCA TACAAGA-3' |
| Antibody | mouse anti-Hsc70 | Enzo Life Sciences | cat# ALX-804–067; RRID: AB_10538284 | 1:500 dilution in SuperBlock, O/N at 4 °C |
| Antibody | mouse anti-Tsg101 | Santa Cruz Biotechnology | cat# sc-7964; RRID: AB_671392 | 1:500 dilution in SuperBlock, O/N at 4 °C |
| Antibody | rabbit anti-Cd63 | System Biosciences | cat# EXOAB-CD63A-1; RRID:AB_2561274 | 1:1000 dilution in SuperBlock, O/N at 4 °C |
| Antibody | mouse anti-Met | Cell Signaling Technology | cat# 3127; RRID:AB_331361 | 1:1000 dilution in SuperBlock, O/N at 4 °C |
| Antibody | rabbit anti-pMet (Tyr 1234/5) | Cell Signaling Technology | cat# 3077; RRID:AB_2315156 | 1:1000 dilution in SuperBlock, O/N at 4 °C |
| Antibody | rabbit anti-Gapdh | Santa Cruz Biotechnology | cat# sc-25778; RRID:AB_10167668 | 1:1000 dilution in SuperBlock, O/N at 4 °C |
| Antibody | rabbit anti-pMet (Tyr 1234/5) | Thermo Fisher Scientific | cat# MA5-15083; RRID:AB_10983015 | 1:1000 dilution in SuperBlock, O/N at 4 °C |
| Antibody | HRP-conjugated sheep anti-mouse | GE Healthcare | cat# NA931; RRID:AB_772210 | 1:15,000 dilution in SuperBlock, 1 hr at RT |
| Antibody | HRP-conjugated donkey anti-rabbit | GE Healthcare | cat# NA934; RRID:AB_772206 | 1:15,000 dilution in SuperBlock, 1 hr at RT |
| Sequence-based reagent | shMet primer | this paper | | Forward: 5'-CCAGACTTT TCATACAAGAATA-3' |
| Sequence-based reagent | universal reverse primer | QuantiMir Kit, System Biosciences | cat# RA420A-1 | |
| Sequence-based reagent | U6 primer | LncRNA Profiler qPCR Array Kit, System Biosciences | cat# RA900A-1 | |
| Sequence-based reagent | *Met* primer | this paper | | Forward: 5'-CTGGACA GATTGTGGGAGTAAG −3' |
| Sequence-based reagent | *Gapdh* primer | LncRNA Profiler qPCR Array Kit, System Biosciences | cat# RA900A-1 | |
| Software, algorithm | ImageJ | doi:10.1038/ nmeth.2089 | RRID:SCR_003070 | version 1.49 u |
| Software, algorithm | SDS | Applied Biosystems | RRID:SCR_015806 | version 2.4 |
| Software, algorithm | Nanoparticle tracking Analysis (NTA) | Nanosight | RRID:SCR_014239 | version 2.3 |
| Software, algorithm | Living Image | Caliper Life Sciences | RRID:SCR_014247 | version 4.4 |

*Continued on next page*

*Continued*

| Reagent type (species) or resource | Designation | Source or reference | Identifiers | Additional information |
|---|---|---|---|---|
| Software, algorithm | R Project for statistical computing | https://www.r-project.org | RRID:SCR_001905 | version 3.5.1 |

As described in the Registered Report (*Lesnik et al., 2016*), we attempted a replication of the experiments reported in Figure 4E and Supplementary Figures 1C and 5A of *Peinado et al., 2012*. A detailed description of all protocols can be found in the Registered Report (*Lesnik et al., 2016*) and are described below with additional information not listed in the Registered Report, but needed during experimentation.

## Cell culture

B16-F10 (RRID:CVCL_0159) and B16-F10-luciferase (expressing firefly luciferase) (shared by Dr. David Lyden, Weill Medical College of Cornell University) were maintained in DMEM (Thermo Fisher Scientific, cat# MT10013CV) supplemented with 10% exosome-depleted fetal bovine serum (FBS) and 100 U/ml penicillin-streptomycin (Thermo Fisher Scientific, cat# 15-140-122) at 37°C in a humidified atmosphere at 5% $CO_2$. FBS (Hyclone, cat# SH30088.03) was depleted of bovine exosomes by ultracentrifugation (Beckman Coulter, Optima L-90K) at 100,000x$g$ (Beckman Coulter, Type 70 Ti Rotor) for 70 min at 4°C. Quality control data for both cell lines are available at https://osf.io/3x58z/. This includes results confirming the cell lines were free of mycoplasma contamination as well as STR DNA profiling of the cell lines (B16-F10 cells: DDC Medical, Fairfield, Ohio; B16-F10-luciferase cells: IDEXX BioResearch, Columbia, Missouri), which were confirmed to be the indicated cell lines when queried against STR profile databases. Additionally, B16-F10-luciferase cells were confirmed free of common mouse pathogens (IDEXX BioResearch).

## Stable cell generation

B16-F10 cells ($2.5\times10^4$ cells/well of 24 well plate) grown in culture medium supplemented with 1X TransDux were transduced with either pGIPZ mouse Met shRNA lentiviral particles (Dharmacon/GE Life Sciences, cat# VGM5520-200377256, clone ID: V3LMM_456078, lot# V16042101, sense sequence: 5'-CCAGACTTTTCATACAAGA-3') or pGIPZ non-silencing shRNA lentiviral control particles (Dharmacon/GE Life Sciences, cat# RHS4348, lot# 150529606, sense sequence: 5'-ATCTCGCTTGGGCGAGAGTAAG-3') at MOIs of 10, 20, or 50, incubated overnight (16 hr), then growth medium was replaced. Transduction efficiency was measured by GFP expression 72 hr later and then transduced cells were selected with 1.5 μg/ml puromycin for 28 days. Cells resistant to puromycin were checked for GFP expression. Growth rates were similar among all stable cell populations generated based on qualitative observations documented during passaging. Cells were frozen down and stored for further use. Thawed cells were maintained in 1.5 μg/ml puromycin, except when cells were plated for experiments. Microscopy images of cells including laboratory notes on cell culturing of stable cell lines are available at https://osf.io/yp5g6/.

## Exosome purification and tracking analysis

Supernatant from B16-F10 shMet and B16-F10 shScr cells collected 48–72 hr after initial plating (two 15 cm plates at ~$5\times10^6$ cells/plate) were pelleted at 500x$g$ (Beckman Coulter, Allegra X-14R) for 10 min at 4°C. The supernatant was clarified by ultracentrifugation (Beckman Coulter, Optima L-90K) at 20,000x$g$ (Beckman Coulter, Type 70 Ti Rotor) for 20 min at 4°C. Exosomes were then collected by ultracentrifugation at 100,000x$g$ for 70 min at 4°C. Protein concentration of exosomes were determined with a BCA assay using a standard curve (data available at https://osf.io/9xnbf/) and either suspended in PBS and used directly for experiments (*in vivo* animal experiments) or stored at −20°C until analysis (Western blots and Nanosight). Characterization of exosomes was performed with a nanoparticle analysis system (Nanosight, LM10) equipped with a blue laser (405 nm) and Nanoparticle tracking Analysis (NTA) software (RRID:SCR_014239), version 2.3. Summary of exosome number and size distribution, with or without the finite track length adjustment (FTLA) algorithm, are provided in *Table 1* as averaged values among the multiple independent exosome preparations

generated during this study. Span was calculated using the formula: Span = (D90-D10)/D50, where D90 is the point in the size distribution where 90% of the sample is contained, D10 is where 10% of the sample is contained, and D50 is where 50% of the sample is contained (i.e. median). Batch analysis reports, including short video clips, are available at https://osf.io/vczn7/ for exosomes purified for *in vitro* experiments and https://osf.io/9xnbf/ for exosomes purified for *in vivo* experiments.

## Western blots

Cell lysate was generated from B16-F10 shMet and B16-F10 shScr cells by removing medium from plates, washing 2X with 1X PBS, followed by addition of 1X RIPA buffer (Sigma-Aldrich, cat# R0278) supplemented with protease inhibitors (Sigma-Aldrich, cat# 4693116001) and phosphatase inhibitors (Sigma-Aldrich, cat# P5726) at manufacturer recommended concentrations. Lysed cells were scraped from plates, incubated on ice for 15 min followed by transfer to −80°C. Cell lysates were thawed on ice, then centrifuged at 12,000x*g* at 4°C before protein concentration of the supernatant was quantified using a BCA protein assay kit following manufacturer's instructions. Previously frozen purified exosomes used for Western blots were resuspended in 1X RIPA buffer supplemented with protease and phosphatase inhibitors. Laemmli sample buffer was added to the purified exosomes or cell lysates and incubated at 95°C for 5 min. 30 μg of purified exosome or cell lysates (or 10 μg of purified exosomes for gels examining exosome markers), along with a protein ladder (Bio-Rad Laboratories, cat# 161–0377), was resolved by SDS-PAGE and transferred to PVDF membrane as described in the Registered Report (*Lesnik et al., 2016*). Membranes were blocked with SuperBlock (Thermo Fisher Scientific, cat# 37516) following manufacturer's instructions. Membranes were cut at ~75 kDa to allow for parallel probing. Incubation with primary antibody, diluted in SuperBlock, was conducted overnight at 4°C. Membranes were probed with: mouse anti-Hsc70 (Enzo Life Sciences, cat# ALX-804–067, RRID:AB_10538284), 1:500 dilution; mouse anti-Tsg101 (Santa Cruz Biotechnology, cat# sc-7964, RRID:AB_671392), 1:500 dilution; rabbit anti-Cd63 (System Biosciences, cat# EXOAB-CD63A-1, RRID:AB_2561274), 1:1000 dilution; mouse anti-Met (Cell Signaling Technology, cat# 3127, RRID:AB_331361), 1:1000 dilution; rabbit anti-pMet (Tyr 1234/5) (Cell Signaling Technology, cat# 3077, RRID:AB_2315156), 1:1000 dilution; rabbit anti-Gapdh (Santa Cruz Biotechnology, cat# sc-25778, RRID:AB_10167668), 1:1000 dilution. Incubations were followed by washes with 1X TBS supplemented with 0.1% tween (TBST) and then the appropriate secondary antibody diluted in SuperBlock for 1 hr at room temperature: HRP-conjugated sheep anti-mouse (GE Healthcare, cat# NA931, RRID:AB_772210), 1:15,000 dilution; HRP-conjugated donkey anti-rabbit (GE Healthcare, cat# NA934, RRID:AB_772206), 1:15,000 dilution. Membranes were washed with TBST and incubated with SuperSignal West Femto Maximum Sensitivity Substrate (Thermo Fisher Scientific, cat# 34095) according to the manufacturer's instructions. Scanned Western blots were quantified using ImageJ software (RRID:SCR_003070), version 1.49u (*Schneider et al., 2012*). We also used an additional antibody targeting pMet (rabbit anti-pMet (Tyr 1234/5) (Thermo Fisher Scientific, cat# MA5-15083, RRID:AB_10983015), 1:1000 dilution) due to an inability to detect a reliable Western blot signal in exosomes with the pMet antibody (RRID:AB_331361) stated in the Registered Report and used in the original study. Both antibodies gave similar results (RRID:AB_10983015: https://osf.io/96tcs/; RRID:AB_331361: *Figure 1B*). Additional method details and image data are available at https://osf.io/aqm2m/.

## Quantitative PCR

Total RNA was isolated from B16-F10 shMet and B16-F10 shScr cells using TRIzol reagent (Thermo Fisher Scientific, cat# 15596026) according to manufacturer's instructions. Total RNA was reverse transcribed into cDNA using a QuantiMir Kit (System Biosciences, cat# RA420A-1) according to manufacturer's instructions. To detect the shMet shRNA expression a primer against the shMet sequence (Forward: 5'-CCAGACTTTTCATACAAGAATA-3') and a universal reverse primer (QuantiMir Kit, System Biosciences, cat# RA420A-1) were used along with a U6 specific primer (LncRNA Profiler qPCR Array Kit, System Biosciences, cat# RA900A-1) and universal reverse primer that were used for normalization. To detect *Met* expression a *Met* specific primer (Forward: 5'-CTGGACAGATTGTGGGAG TAAG −3') and universal reverse primer were used along with a *Gapdh* specific primer (LncRNA Profiler qPCR Array Kit) and universal reverse primer that were used for normalization. qRT-PCR reactions were performed in technical triplicate with Power SYBR Green PCR Master Mix (Thermo Fisher

Scientific, cat# 4368577) according to manufacturer's instructions. PCR cycling conditions were [1 cycle 50°C for 2 min, 95°C for 10 min – 40 cycles 95°C for 15 s, 60°C for 60 s – dissociation stage: 95°C for 15 s, 60°C for 15 s] using an Applied Biosystems real-time PCR system (Applied Biosystems, 7900 HT Fast Real-Time PCR System) and SDS software (RRID:SCR_015806), version 2.4. Negative controls containing no cDNA template were included. Relative expression levels were determined using the ΔΔCt method.

## Animals

All animal procedures were approved by the Stanford University IACUC# 30226 and were in accordance with the Stanford University policies on the care, welfare, and treatment of laboratory animals. No blinding occurred during the experiments.

Six-week old female C57BL/6 mice (Charles River, Strain code: 027, RRID:IMSR_CRL:27) were housed in sterile conditions under standard temperature, humidity, and timed lighting conditions with 12 hr light/dark cycles, and were provided with sterile rodent chow and water *ad libitum*. Mice were randomized and injected, via retro-orbital and alternating injections between left and right eyes, three times a week for a total of 28 days with either 5 µg of freshly-isolated B16-F10 shScr exosomes, 5 µg of freshly-isolated B16-F10 shMet exosomes, or 1.25 µg synthetic 100 nm liposomes (Encapsula NanoSciences) (mimics 5 µg exosome protein, based on a theoretical 4:1 protein:L-α-phosphatidylcholine ratio) in 100 µl filtered Phosphate Buffered Saline (PBS) (Thermo Fisher Scientific, cat# MT21040CM). The exosomes were resuspended right before each individual injection by pipetting 3–4 times, tapping the tube, inverted 3–4 times, and then tapping the syringe right before each injection. After exosome injections (12 total), mice were inoculated subcutaneously (s.c.) with $1 \times 10^6$ B16-F10-luciferase cells in 100 µl of PBS in the dorsal area. Primary tumor (caliper measurements) and body weight were measured three times a week. The planned study design involved waiting 21 days after B16-F10-luciferase tumor cell implantation before sacrificing the mice for analysis; however, two animals were found dead (one injected with B16-F19 shScr exosomes and one injected with synthetic liposomes) before this time point was reached (17 days after implantation), which in addition the largest tumors of the surviving mice having reached >1000 mm$^3$, prompted us to stop the experiment early (18 days after implantation). To measure metastasis, mice were anesthetized (using isoflurane and $O_2$) and 50 mg/kg of D-luciferin in 50–100 µl PBS was injected retro-orbitally. Five minutes later, mice were euthanized by cervical dislocation and primary tumors and organs (femurs and lungs) were dissected. Primary tumors were weighed (Delta Range scale: Metler Toledo, Model # PB303-S) and organs (femurs and lungs) were analyzed for luciferase expression using the IVIS Spectrum system (Caliper, Xenogen). Anesthesia, luciferin injections, euthanasia, dissection, and imaging/weighing were performed with mice from different groups in parallel (i.e. one from each of the three groups) so variations during the procedure were equal across groups.

## IVIS imaging

Images were acquired in a Xenogen IVIS Spectrum at a medium binning level (8) and a 22.6 cm field of view. Acquisition times were set to auto-exposure and were required at 1 and 4 min. There was high concordance between the two exposures (correlation coefficient ($\rho$)=0.97). The 4 min exposure is shown in figures and used in the statistical analysis. Additionally, after this imaging was complete, the organs were soaked for 10–15 min in 4 ml of 50 mg/ml D-luciferin in six well tissue culture plates and imaged again at 1 and 3 min, which had high concordance with the first set of images taken. Living Image software (RRID:SCR_014247), version 4.4, was used for quantitative analysis. Image files are available at https://osf.io/9xnbf/.

## Statistical analysis

Statistical analysis was performed with R software (RRID:SCR_001905), version 3.5.1 (*Core TeamR, 2018*). All data, csv files, and analysis scripts are available on the OSF (https://osf.io/ewqzf/). Confirmatory statistical analysis was pre-registered (https://osf.io/g5tzn/) before the experimental work began as outlined in the Registered Report (*Lesnik et al., 2016*). Data were checked to ensure assumptions of statistical tests were met and in the case of the metastatic data were natural log transformed to achieve a normal distribution and equal variance while also allowing for comparisons on a modified percentage scale (*Cole and Altman, 2017*). When described in the results, the

Bonferroni correction, to account for multiple testings, was applied to the alpha error or the *p*-value. The Bonferroni corrected value was determined by divided the uncorrected value (0.05) by the number of tests performed. A meta-analysis of a common original and replication effect size was performed with a random effects model and the *metafor* R package (*Viechtbauer, 2010*) (https://osf.io/c69jx/). Meta-analyses were performed without weighting, since unweighted Cliff's *d* has been reported to reduce bias (*Kromrey et al., 2005*). The asymmetric confidence intervals for the overall Cliff's *d* estimate was determined using the normal deviate corresponding to the (1 - alpha/2)[th] percentile of the normal distribution (*Cliff, 1993*). The raw data pertaining to Figure 4E and Supplementary Figure 5A of *Peinado et al., 2012* were shared by the original authors. The summary data was published in the Registered Report (*Lesnik et al., 2016*) and used in the power calculations to determine the sample size for this study.

## Data availability

Additional detailed experimental notes, data, and analysis are available on OSF (RRID:SCR_003238) (https://osf.io/ewqzf/; *Kim et al., 2018*). This includes the R Markdown file (https://osf.io/hz3k7/) that was used to compose this manuscript, which is a reproducible document linking the results in the article directly to the data and code that produced them (*Hartgerink, 2017*).

## Deviations from registered report

We planned to generated B16-F10 shMet and shScr stable cells with an MOI of 10, similar to the original study, but observed that the Met levels in the shScr cells at this MOI were, for unknown reasons, low when compared to the shScr cells generated at the other MOI ratios (*Figure 1—figure supplement 1C*). Experiments reported in this study were with stable cells generated with an MOI of 20. We also included additional tests (qRT-PCR) to confirm the presence of shMet, and corresponding decrease in Met expression, in the stable cells generated. The cell lysis buffer was additionally supplemented with phosphatase inhibitors in an attempt to increase detection of pMet. We also tried an additional antibody targeted against pMet (RRID:AB_10983015) due to an inability to detect a reliable Western blot signal in exosomes with the p-Met antibody stated in the Registered Report and used in the original study (RRID:AB_2315156). Both antibodies gave similar results (RRID:AB_10983015: https://osf.io/96tcs/; RRID:AB_331361: *Figure 1B*). For the *in vivo* experimentation, the planned study design involved waiting 21 days after B16-F10-luciferase tumor cell implantation before sacrificing the mice for analysis; however, two animals were found dead (one injected with B16-F19 shScr exosomes and one injected with synthetic liposomes) before this time point was reached, which in addition the largest tumors of the surviving mice having reached >1000 mm$^3$, prompted us to stop the experiment early (18 days after implantation). Additional materials and instrumentation not listed in the Registered Report, but needed during experimentation are also listed.

## Acknowledgements

The Reproducibility Project: Cancer Biology would like to thank Dr. David Lyden (Weill Medical College of Cornell University) and Hector Peinado (Spanish National Cancer Research Centre) for sharing critical reagents, data, and protocol information during preparation of the Registered Report, specifically the B16-F10 and B16-F10-luciferase cells. We would also like to thank Courtney Soderberg at the Center for Open Science for assistance with statistical analyses, Jacob Lesnik for managing resources and collaboration details at System Biosciences LLC, the Stanford Transgenic, Knockout and Tumor model Center at the Stanford Cancer Institute, and the following companies for generously donating reagents to the Reproducibility Project: Cancer Biology; American Type and Tissue Collection (ATCC), Applied Biological Materials, BioLegend, Charles River Laboratories, Corning Incorporated, DDC Medical, EMD Millipore, Harlan Laboratories, LI-COR Biosciences, Mirus Bio, Novus Biologicals, Sigma-Aldrich, and System Biosciences (SBI).

# Additional information

## Group author details

**Reproducibility Project: Cancer Biology**
**Elizabeth Iorns**: Science Exchange, Palo Alto, United States; **Rachel Tsui**: Science Exchange, Palo Alto, United States; **Alexandria Denis**: Center for Open Science, Charlottesville, United States; **Nicole Perfito**: Science Exchange, Palo Alto, United States; **Timothy M Errington**: Center for Open Science, Charlottesville, United States; **Elizabeth Iorns**: Science Exchange, Palo Alto, United States; **Rachel Tsui**: Science Exchange, Palo Alto, United States; **Alexandria Denis**: Center for Open Science, Charlottesville, United States; **Nicole Perfito**: Science Exchange, Palo Alto, United States; **Timothy M Errington**: Center for Open Science, Charlottesville, United States

## Competing interests

Jeewon Kim, Vittorio Sebastiano: Stanford Transgenic, Knockout and Tumor model Center is a Science Exchange associated lab. Amirali Afshari, Ranjita Sengupta, Archana Gupta, Young H Kim: System Biosciences LLC is a Science Exchange associated lab and offers products and services for exosome research. Exosome detection products, specifically exosome specific antibodies, and nanoparticle tracking analysis (NTA) service were used during this study; however, ultracentrifugation-free exosome isolation products and services were not used. Reproducibility Project: Cancer Biology: EI, RT, NP: Employed by and hold shares in Science Exchange IncThe other authors declare that no competing interests exist.

## Funding

| Funder | Author |
| --- | --- |
| Laura and John Arnold Foundation | Reproducibility Project: Cancer Biology |

The funder had no role in study design, data collection and interpretation, or the decision to submit the work for publication.

## Author contributions

Jeewon Kim, Acquisition of data, Analysis and interpretation of data, Drafting or revising the article, Performed all exosome purification and generation of exosome-free FBS, qPCR, Animal experiments and IVIS imaging; Amirali Afshari, Ranjita Sengupta, Archana Gupta, Young H Kim, Acquisition of data, Analysis and interpretation of data, Drafting or revising the article, Performed stable cell line generation, Western blots, qPCR, Nanosight analysis; Vittorio Sebastiano, Reproducibility Project: Cancer Biology, Analysis and interpretation of data, Drafting or revising the article

## Author ORCIDs

Alexandria Denis http://orcid.org/0000-0002-1210-2309
Timothy M Errington http://orcid.org/0000-0002-4959-5143
Alexandria Denis http://orcid.org/0000-0002-1210-2309
Timothy M Errington http://orcid.org/0000-0002-4959-5143

## Ethics

Animal experimentation: All animal procedures were approved by the Stanford University IACUC# 30226 and were in accordance with the Stanford University policies on the care, welfare, and treatment of laboratory animals.

## Decision letter and Author response

Decision letter https://doi.org/10.7554/eLife.39944.014
Author response https://doi.org/10.7554/eLife.39944.015

# Additional files

## Supplementary files

• Transparent reporting form
DOI: https://doi.org/10.7554/eLife.39944.008
• Reporting standard 1 The ARRIVE guidelines checklist.
DOI: https://doi.org/10.7554/eLife.39944.009

## Data availability

Additional detailed experimental notes, data, and analysis are available on OSF (RRID:SCR_003238) (https://osf.io/ewqzf/; Kim et al., 2018). This includes the R Markdown file (https://osf.io/hz3k7/) that was used to compose this manuscript, which is a reproducible document linking the results in the article directly to the data and code that produced them (Hartgerink, 2017).

The following dataset was generated:

| Author(s) | Year | Dataset title | Dataset URL | Database and Identifier |
|---|---|---|---|---|
| Kim J, Afshari A, Sengupta R, Sebastiano V, Gupta A, Kim YH, Iorns E, Tsui R, Denis A, Perfito N, Errington TM | 2018 | Study 42: Replication of Peinado et al., 2012 (Nature Medicine) | http://dx.doi.org/10.17605/OSF.IO/EWQZF | Open Science Framework, 10.17605/OSF.IO/EWQZF |

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
