## [Decision Letter]

Thank you for submitting your work entitled "Replication Study: Melanoma exosomes educate bone marrow progenitor cells toward a pro-metastatic phenotype through MET" for consideration at *eLife*. Your article has been evaluated by Charles Sawyers as the Senior Editor, Michael Green as the Reviewing Editor, and two expert reviewers.

There are a number of issues that need to be seriously addressed as outlined below in the separate reviews. We realise that additional work that was not outlined in the Registered Report is outside the scope of the project, but we would ask you to respond to each point, revising the text accordingly, and stating the necessary caveats in response (rather than performing additional experiments).

We would like to emphasize that in addition to changes in the main text, the Abstract must be modified to accurately reflect the data generated and their issues. Reviewer 1 had this to add about the Abstract:

"Most importantly, the abstract needs to be revised to specifically state the issues with animal death, failure of tumors to grow and off target effects of the control encountered in this report but not the original study. Moreover, the Abstract should clearly state that any lack of reproducibility is due to the aforementioned issues. Specifically, it should state that the failure to reproduce the results of the original study is because the author's own controls and experiments failed technically. Thus, no conclusions can be drawn."

Reviewer #1:

We would like to thank the authors for transparency in reporting results and analyses. However, upon reviewing the report, we identified several concerns that make it impossible to draw definitive conclusions from the presented data. Therefore, the study should not be published as is, and suggest that the authors perform additional work to address the major problems they have encountered that are not otherwise reported in other studies in the field, including the original study. Moreover, the authors need to include additional controls that would allow direct comparison with previous studies.

First and foremost, the Abstract, as written, is misleading and should be re-written before publication is considered. In the current version, the Abstract states that the authors 1) could not "reliably detect phospho-Met (pMet) in exosomes" and that 2) and that "while the effects [on metastasis] were in the same direction as the original study (Figure 4E; Peinado et al., 2012), they were not statistically significant". These conclusions cannot be drawn based on the data presented in the report due to experimental design and technical issues detailed below. Therefore, the statements in the Abstract need to be amended to reflect the measurement variability, lack of tumor growth in some animals, animal death and off target effects evident in this report.

The most significant issue encountered in this report is the fact that the authors' data on both Met expression and primary tumor growth indicates there are off-target effect(s) with their "non-targeting" scrambled shRNA. A more appropriate control, at least done in parallel, would have been to use the shEmpty vector that does not contain a scrambled sequence. An even better and more relevant control would have been to use the unmodified/untransduced, parental B16F10 cell line-derived exosomes for education and ensure that a level of primary tumor growth and metastasis equivalent to the original study is obtained under these conditions.

Specifically, in Figure 1A, the shScr control cells should express Met, while the shMet cells should not express Met. The data presented by the authors shows the contrary – is this a labeling mistake? Moreover, it is unclear from the data presented in Figure 1A and B if the replicates are technical or biological replicates (if so, are they independent exosome collections from the same batch of shScr or shMet lentivirus-infected cells, or are they independent infections?).

Unfortunately, it is impossible to determine if the authors' inability to detect pMet in shScr and scMet exosomes is a technical issue or a problem with the cell models, because the essential control B16F10 cells and exosomes that should have robust expression of Met and pMet was not used. The authors need to revise their experimental design and repeat their experiments including this control, which was present in the original Peinado et al. paper as well as in numerous other publications before any conclusions can be drawn. Importantly, the authors should highlight that, consistent with original report by Peinado et al., the shMet construct efficiently downregulated Met in B16F10 cells and their exosomes.

There could be several explanations that need to be explored that could account for the lack of pMet detection in exosomes in this report. First of all, it is unclear if fresh exosomes had been used in these Western Blot experiments (authors only indicate the use of freshly-isolated exosomes in in vivo studies) but it would also be critical that fresh exosomes be used for phospho-protein detection in exosomes. The authors actually state that exosomes were "stored at -20°C until analysis” therefore, the analysis of phosphoproteins should be repeated immediately following isolation, resuspension in PBS, measurement of protein and resuspension in Laemmli buffer. This should all be performed using fresh exosomes.

Importantly, many published studies, in additional to the original report from Peinado et al., 2012, have since validated and were able to detect Met as well as pMet in B16F10 exosomes and other exosomes. These studies are listed below, they should be included in the reference list of this report and discussed in the context of the authors' results.

• Steenbeek et al., 2018 – Figure 4 shows Met expression in B16F10 exosomes

• Barrow-McGee R et al., Nature Communications, PMID: 27336951 – shows cMet and pMet in endosomes throughout the paper

• Tripolitsioti D et al., Oncotarget, PMID: 29796184 – throughout the paper and Supplementary Figure 5D (Met), 6B (pMet)

• Adachi et al., 2016 – shows both Met and pMet in Figure 1C

• Cannistraci et al., 2017 – shows both Met and pMet in Figure 2D

• Plebanek MP et al., 2017

• Zeng et al., 2017 – shows both Met and pMet

• He et al., 2015 – shows both Met and pMet in Figure 4E.

Another cause for concern with this report is the off target effect present in the only control used in this study, the shScr control. As shown in Figure 2A of the report and pointed out by the authors in their results, using a lower multiplicity of infection (MOI) of 10, the same used in the original Peinado et al. paper, the authors of this report find there is a significant on target effect of the control shScr, as Met levels are downregulated. Strangely, the authors are able to reduce the on target effect of the control shScr by increasing the MOI to 20 or 50. Most likely, this is due to the oligomers binding to other targets, and thus an increase in off target effects. Since these are commercial lentiviral particles, and the time passed since the initial report, it would be critical for the authors to verify with the company that they still use the same sequence as control as reported in the original paper. In this manuscript the authors do not define the targeting sequence as in the original report (Peinado et al., specified that sense sequence used, obtained from Thermo was: 5′-ATCTCGCTTGGGCGAGAGTAAG-3′) but the authors in the current report do not describe the sequence and use pGIPZ non-silencing shRNA lentiviral control particles from Dharmacon/GE Life 281 Sciences, cat# RHS4348, lot# 150529606. Moreover, the difference in results could be due to lot variability or quality of commercial virus (this should be discussed by the authors in the report). Minor concerns regarding Figure 2 are the inconsistency of GAPDH loading control levels as well as the presentation of MOIs out of order (10, then 50, then 20). The authors need to provide empty vector and non-transduced cell controls for these experiments. Moreover, to investigate the off target effects of the shScr they should perform transcriptomics analyses of the non-transduced, empty vector, shScr and shMet B16-F10 cells. They can also perform unbiased proteomics analysis of shScr versus shMet versus B16F10 exosomes to determine if/how the cargo of shScr exosomes is affected by off target effects. The authors should test proliferation and apoptosis of shScr cells, to account for any consequences of off target effects of the shScr construct on the health of these cells. This is critical because of the issues evident in Figure 3.

The other major cause of concern, that precludes the data from this report being interpretable and the drawing of any conclusions, is, as evident from Figure 3A, B), the fact that the primary tumors growing in animals educated with shScr exosomes are smaller than those in the liposome-treated controls, could this be because of their issues with the control particles? Additionally, whereas the size of primary tumors growing in the liposome or shMET treated groups is reproducible and similar, there is variability in the growth of the tumors in the shScr treated mice, with some of the tumors as small as 0.025 g, which is 32-fold smaller than the mean of the other groups, which is 0.8 g. Consistent with the original report by Peinado et al., the authors do not find differences in tumor growth between liposome treated mice and shMet mice. Given the dramatic reduction in tumor growth in the shScr control treated mice, and the large variability in the growth of these tumors, it is not surprising that the authors of this report fail to find statistically significant differences in lung and bone metastasis between the shScr control and shMet exosome educated mice. Therefore, before any conclusions can be drawn, and to account for the large variability and off target effect of the shScr control and in this experiment overall, the authors need to repeat the experiment increasing animal numbers and including animals educated with un-transduced, parental B16F10 exosomes to demonstrate that, in their hands, they can obtain consistent primary tumor growth and metastasis levels without the confounding contribution of the off-target effects in the shScr. Moreover, to reiterate, the differences between the shScr and shMet cannot be deemed non-significant since the lack of significance stems from the high variance and lower than expected primary tumor size in the shScr control group.

Importantly, there is lack of tumor growth in some animals and death of animals injected with exosomes. B16F10 are aggressive tumors with robust growth that do not fail to implant or grow, therefore the fact that the authors of this study encounter challenges in having tumors grow in all animals injected with control cells reflect either 1) technical issues with cell viability, injection, etc. and/or 2) the fact that the off target effects of the shScr exosomes used for education are effectively inhibiting tumor growth. These issues need to be resolved before this report can be published.

Additionally, it is concerning that education of animals with 5 μg of shScr B16F10 exosomes leads to the death of some animals. This was not reported in publications with either 5 μg or 10 μg of B16F10 exosomes (for example in the original publication or in Dr. Olga Volpert's Nature Communications, 2018 report injected 10 μg of B16F10 exosomes repeatedly with no death). The deaths observed by the authors of the present study upon exosome injection are worrisome. Have the authors verified that they are measuring exosomal protein accurately after isolation? The authors should define the buffer used for exosome resuspension (in the original report was PBS as stated "The floating exosome fraction was collected again by ultracentrifugation as above, and the final pellet was resuspended in PBS"). Mouse lethality could reflect inaccurate exosome protein quantification. Alternatively, the presence of exosome aggregates in the preparation would lead to animal death, to avoid this exosomes should be vigorously resuspended before injection (10 times at least). Last but not least, the health of the cells from which the exosomes are collected can significantly influence the quality of the exosomes and could affect animal health (especially in the context of the shScr off target effects).

Reviewer #2:

Overall this replication study was carried out appropriately. I have some concerns about the quality of the data presented (in point 1) and some other comments, as follows:

1) In Figure 1, The Western blots give a variety of concerns. First, they are all very dark and overblown. I would also like to see the full blots – they should probably be included in the published paper. Second, for Figure 1A, the blot comparing the shScr and shMet in the cells shows much higher Met in the shMet cells then in the controls. I presume that is a mistake? Third, is GAPDH a relevant loading control or does it change in exosomes with Met?

2) For Figure 1—figure supplement 1, I also don't see how a shMet can be compared to shSc when they are on different blots and it seems strange to compare the MOI to just each other and not untreated cells. This is a relatively minor concern – the assessment of Met KD in Figure 1 is the most important.

3) For the analysis of tumor metastasis, are the data normal or not normal? It seems like statistical analyses were used that are appropriate for data with a non-Gaussian distribution. Therefore, I'm not sure why mean and SEM are used to represent the data, which are only appropriate for data with a Gaussian distribution.

4) For differences between this study and Peinado that are listed in the last paragraph of the Results/Discussion section of the paper, I think differences in Met signaling and in the effectiveness of knockdown (which I also doubt based on the quality of the Western blots here) should be mentioned. In addition, the study was not carried out as long as the Peinado study (18 days instead of 21 days) after exosome education, so this should be mentioned in the last paragraph too. There were also fewer mice, which could definitely affect detection of statistical significance, which should be mentioned.

---

## [Author Response]

There are a number of issues that need to be seriously addressed as outlined below in the separate reviews. We realise that additional work that was not outlined in the Registered Report is outside the scope of the project, but we would ask you to respond to each point, revising the text accordingly, and stating the necessary caveats in response (rather than performing additional experiments).We would like to emphasize that in addition to changes in the main text, the Abstract must be modified to accurately reflect the data generated and their issues. Reviewer 1 had this to add about the Abstract:"Most importantly, the abstract needs to be revised to specifically state the issues with animal death, failure of tumors to grow and off target effects of the control encountered in this report but not the original study. Moreover, the Abstract should clearly state that any lack of reproducibility is due to the aforementioned issues. Specifically, it should state that the failure to reproduce the results of the original study is because the author's own controls and experiments failed technically. Thus, no conclusions can be drawn."

We disagree with the statement that ‘failure to reproduce the results of the original study is because the author’s own controls and experiments failed technically’. The replication was performed with the same controls, cells, shRNA sequences, etc. as reported in the original study for these experiments with input from the original authors and peer review of the Registered Report. The specific concerns raised have been addressed below and as appropriately the text has been revised. Additionally, while technical concerns and hidden moderators might be factors that influence both the original and replication results, as Reviewer #1 suggests, another likely factor is biological variation. We agree that the results we reported could be attributed to observable, or hypothetical, differences between the original study and this replication and we have revised the Abstract to highlight this.

Reviewer #1:We would like to thank the authors for transparency in reporting results and analyses. However, upon reviewing the report, we identified several concerns that make it impossible to draw definitive conclusions from the presented data. Therefore, the study should not be published as is, and suggest that the authors perform additional work to address the major problems they have encountered that are not otherwise reported in other studies in the field, including the original study. Moreover, the authors need to include additional controls that would allow direct comparison with previous studies.First and foremost, the Abstract, as written, is misleading and should be re-written before publication is considered. In the current version, the Abstract states that the authors 1) could not "reliably detect phospho-Met (pMet) in exosomes" and that 2) and that "while the effects [on metastasis] were in the same direction as the original study (Figure 4E; Peinado et al., 2012), they were not statistically significant". These conclusions cannot be drawn based on the data presented in the report due to experimental design and technical issues detailed below. Therefore, the statements in the Abstract need to be amended to reflect the measurement variability, lack of tumor growth in some animals, animal death and off target effects evident in this report.The most significant issue encountered in this report is the fact that the authors' data on both Met expression and primary tumor growth indicates there are off-target effect(s) with their "non-targeting" scrambled shRNA. A more appropriate control, at least done in parallel, would have been to use the shEmpty vector that does not contain a scrambled sequence. An even better and more relevant control would have been to use the unmodified/untransduced, parental B16F10 cell line-derived exosomes for education and ensure that a level of primary tumor growth and metastasis equivalent to the original study is obtained under these conditions.

The ‘non-targeting’ scrambled shRNA used in this replication was the same control used in the original study per the original paper methods and our communication with the original authors. The same sequence was confirmed from the commercial supplier before the replication was conducted and we included additional details of the sequence in the revised manuscript.

We agree that inclusion of untransduced B16-F10 cell line-derived exosomes would provide insight about possible effects that might have occurred during infection, such as genome instability. We have revised the manuscript to highlight the addition of untransduced B16-F10 cells as a control to consider for future experiments. For the experiments replicated, however, this additional control was not reported in the original study to allow for a comparison between untransduced B16-F10 and shScr cells (Figure 4A of the original study is between B16-F10 and B16-F1 cells and exosomes, Supplementary Figure 5A is between B16-F10, B16-F1, and B16-F10 shMet cells and exosomes, and Figure 4E of the metastasis results is between B16-F10, B16-F10 shMet, and liposome control). It was unclear from the original paper if the B16-F10 cells and exosomes used for the education experiment that was replicated were from untransduced cells or were shScramble; however, during peer review of the Registered Report it was suggested that the B16-F10 cells reported were shScramble transduced. Thus, we conducted this experiment similar to the original experimental design, which was reviewed by the original authors and peer reviewed prior to beginning the experimental work.

Specifically, in Figure 1A, the shScr control cells should express Met, while the shMet cells should not express Met. The data presented by the authors shows the contrary – is this a labeling mistake? Moreover, it is unclear from the data presented in Figure 1A and B if the replicates are technical or biological replicates (if so, are they independent exosome collections from the same batch of shScr or shMet lentivirus-infected cells, or are they independent infections?).Unfortunately, it is impossible to determine if the authors' inability to detect pMet in shScr and scMet exosomes is a technical issue or a problem with the cell models, because the essential control B16F10 cells and exosomes that should have robust expression of Met and pMet was not used. The authors need to revise their experimental design and repeat their experiments including this control, which was present in the original Peinado et al. paper as well as in numerous other publications before any conclusions can be drawn. Importantly, the authors should highlight that, consistent with original report by Peinado et al., the shMet construct efficiently downregulated Met in B16F10 cells and their exosomes.

Figure 1A in the original submission was a labeling mistake that has been fixed in the revised manuscript. Regarding the data presented in Figure 1A and B, these are repeats from ‘independently isolated exosome preparations’ as stated in the figure legend. However, these are from the same batch of shScr or shMet lentivirus-infected cells and assessed after 28 days of puromycin selection as suggested during peer review of the Registered Report. We have added additional details to the revised manuscript to further clarify this point.

As stated above, the suggested additional control (untransduced B16-F10 cells) was not included in the Registered Report, but we have revised the manuscript to suggest the inclusion of this control in future experiments. Importantly, the parental B16-F10 cells (as well the B16-F10-luciferase cells) were shared by the original authors. We have also highlighted that the shMet construct reduced Met expression in B16-F10 cells and exosomes, similar to the original study in the Abstract and at the end of the “Generation and characterization of shMet B16-F10 cells and exosomes” section of the Results. However, the level of knockdown achieved in shMet exosomes was not reported in the original study.

*There could be several explanations that need to be explored that could account for the lack of pMet detection in exosomes in this report. First of all, it is unclear if fresh exosomes had been used in these Western Blot experiments (authors only indicate the use of freshly-isolated exosomes in* in vivo *studies) but it would also be critical that fresh exosomes be used for phospho-protein detection in exosomes. The authors actually state that exosomes were "stored at -20°C until analysis” therefore, the analysis of phosphoproteins should be repeated immediately following isolation, resuspension in PBS, measurement of protein and resuspension in Laemmli buffer. This should all be performed using fresh exosomes.*

We agree there are many possible explanations that could account for the lack of pMet detection in exosomes. As the reviewer highlighted, we did not use fresh exosomes for Western blot analysis (as stated in the Registered Report as well), but did for the education experiment. This is because in our correspondence with the original author for the experiment in question (Protocol 2 of the Registered Report we were told ‘Exosomes for WB can be stored at -20 for 2-3 weeks’.). We agree this is a reasonable explanation, which we have included in the Discussion of the revised manuscript.

Importantly, many published studies, in additional to the original report from Peinado et al., 2012, have since validated and were able to detect Met as well as pMet in B16F10 exosomes and other exosomes. These studies are listed below, they should be included in the reference list of this report and discussed in the context of the authors' results.• Steenbeek et al., 2018 – Figure 4 shows Met expression in B16F10 exosomes• Barrow-McGee R et al., Nature Communications, PMID: 27336951 – shows cMet and pMet in endosomes throughout the paper• Tripolitsioti D et al., Oncotarget, PMID: 29796184 – throughout the paper and Supplementary Figure 5D (Met), 6B (pMet)• Adachi et al., 2016 – shows both Met and pMet in Figure 1C• Cannistraci et al., 2017 – shows both Met and pMet in Figure 2D• Plebanek MP et al., 2017• Zeng et al., 2017 – shows both Met and pMet• He et al., 2015 – shows both Met and pMet in Figure 4E.

We have included additional references in the revised manuscript in addition to the references that were already included on the identification of Met and pMet in exosomes. Some of these references reported the presence of Met in other vesicles and were not included (e.g. Barrow-McGee et al., 2016).

Another cause for concern with this report is the off target effect present in the only control used in this study, the shScr control. As shown in Figure 2A of the report and pointed out by the authors in their results, using a lower multiplicity of infection (MOI) of 10, the same used in the original Peinado et al. paper, the authors of this report find there is a significant on target effect of the control shScr, as Met levels are downregulated. Strangely, the authors are able to reduce the on target effect of the control shScr by increasing the MOI to 20 or 50. Most likely, this is due to the oligomers binding to other targets, and thus an increase in off target effects. Since these are commercial lentiviral particles, and the time passed since the initial report, it would be critical for the authors to verify with the company that they still use the same sequence as control as reported in the original paper. In this manuscript the authors do not define the targeting sequence as in the original report (Peinado et al., specified that sense sequence used, obtained from Thermo was: 5′-ATCTCGCTTGGGCGAGAGTAAG-3′) but the authors in the current report do not describe the sequence and use pGIPZ non-silencing shRNA lentiviral control particles from Dharmacon/GE Life 281 Sciences, cat# RHS4348, lot# 150529606. Moreover, the difference in results could be due to lot variability or quality of commercial virus (this should be discussed by the authors in the report). Minor concerns regarding Figure 2 are the inconsistency of GAPDH loading control levels as well as the presentation of MOIs out of order (10, then 50, then 20). The authors need to provide empty vector and non-transduced cell controls for these experiments. Moreover, to investigate the off target effects of the shScr they should perform transcriptomics analyses of the non-transduced, empty vector, shScr and shMet B16-F10 cells. They can also perform unbiased proteomics analysis of shScr versus shMet versus B16F10 exosomes to determine if/how the cargo of shScr exosomes is affected by off target effects. The authors should test proliferation and apoptosis of shScr cells, to account for any consequences of off target effects of the shScr construct on the health of these cells. This is critical because of the issues evident in Figure 3.

We disagree with the reviewer that an effect on Met expression could be due to the oligomers preferentially targeting Met in the shScr condition. For this to occur it would suggest overexpression of a random sequence would divert the transcriptional/translational machinery away from making Met mRNA/protein or random integration into Met or a gene that regulates Met. Of note, these concerns also exist for the original study since the same oligo sequence was used. It is more likely that non-random genetic/transcriptional drift, which could have been caused by selective pressure from puromycin, affected gene expression of Met (as well as other genes). This has been highlighted as a possible explanation of the results in the revised manuscript.

As mentioned above, the ‘non-targeting’ scrambled shRNA used in this replication was the same control used in the original study per the original paper methods and our communication with the original authors. The same sequence as the original study was confirmed from the commercial supplier before the replication was conducted and we have included additional details on the sequence in the revised manuscript. Additionally, this was shared from the commercial supplier, “this sequence does not match any known mammalian genes and had at least 3 or more mismatches against any gene as determined via nucleotide alignment/BLAST of the 22mer sense sequence”. While the supplier is different, because of company acquisitions that have occurred between the original study and this replication, the vectors used to generate the lentiviral particles were the same as the original study. We agree lot variability is another source that is different between the two studies, although we do not know the lot used in the original study, and we have included this in the revised manuscript.

As mentioned above, the suggested additional control (untransduced B16-F10 cells) was not included in the Registered Report, but we have revised the manuscript to suggest the inclusion of this control in future experiments. Regarding the comment about Gapdh loading, we loaded equal amounts of protein (30 µg) for each sample analyzed by Western blot, opposed to adjusting amounts based on Gapdh levels.

We agree that if there were differences in cell growth, or apoptosis rates could confound the results. This is also true for the original study, which does not report what the growth, or apoptosis rates were for the experiments reported. While we did not perform an experiment to quantify growth or apoptosis rates, we did monitor the health of the cultures and qualitatively kept track of the cell growth rates among the conditions. Growth rates were similar among all stable cell populations generated based on qualitative observations documented during passaging.

The other major cause of concern, that precludes the data from this report being interpretable and the drawing of any conclusions, is, as evident from Figure 3A, B), the fact that the primary tumors growing in animals educated with shScr exosomes are smaller than those in the liposome-treated controls, could this be because of their issues with the control particles? Additionally, whereas the size of primary tumors growing in the liposome or shMET treated groups is reproducible and similar, there is variability in the growth of the tumors in the shScr treated mice, with some of the tumors as small as 0.025 g, which is 32-fold smaller than the mean of the other groups, which is 0.8 g. Consistent with the original report by Peinado et al., the authors do not find differences in tumor growth between liposome treated mice and shMet mice. Given the dramatic reduction in tumor growth in the shScr control treated mice, and the large variability in the growth of these tumors, it is not surprising that the authors of this report fail to find statistically significant differences in lung and bone metastasis between the shScr control and shMet exosome educated mice. Therefore, before any conclusions can be drawn, and to account for the large variability and off target effect of the shScr control and in this experiment overall, the authors need to repeat the experiment increasing animal numbers and including animals educated with un-transduced, parental B16F10 exosomes to demonstrate that, in their hands, they can obtain consistent primary tumor growth and metastasis levels without the confounding contribution of the off-target effects in the shScr. Moreover, to reiterate, the differences between the shScr and shMet cannot be deemed non-significant since the lack of significance stems from the high variance and lower than expected primary tumor size in the shScr control group.

We agree that there was a higher relative standard deviation (RSD) among the results reported in the original study and this replication attempt for the metastatic burden. We included the RSD for both the original and replication results to provide a direct comparison. Importantly, though we cannot compare the primary tumor growth variability as this was not reported in the original study. Furthermore, the variability should also be viewed in the context that cell growth is exponential under an ideal scenario, thus the RSD is reported in natural log units (Cole and Altman, 2017). There are factors that influence and alter the growth of the tumor initially compared to the continued growth of the tumor in vivo, such as availability of nutrients, oxygen, and space. These points have been included in the discussion of the revised manuscript.

This replication attempt, like all of the replication attempts in the Reproducibility Project: Cancer Biology, are designed to perform independent replications with a calculated sample size to detect the originally reported effect size with at least 80% power based on the originally reported data, which was followed in this replication study. We agree the higher variance in the replication is a factor influencing if statistical significance is reached and include this as a consideration when discussing the effect sizes. While technical concerns and hidden moderators might be factors that influence both the original and replication results, as the reviewer suggests, another likely factor is biological variation. That is, one would not expect tumors in individual animals to grow at the exact same rate as there are many influencing factors that cannot be controlled or explained.

We also agree performing another attempt of this experiment with modifications would begin to explore if these, or other, factors influence the outcome of this study. While, it’s not within the scope of this project, or as part of this publishing model, to also conduct these experiments, the results of this replication bring variables not previously thought to influence the experiment into question (size of the control tumors at the end of the study, length of treatment, etc.). Importantly though, it is because of the results that these and other aspects now become targets for hypothesizing and investigation.

Importantly, there is lack of tumor growth in some animals and death of animals injected with exosomes. B16F10 are aggressive tumors with robust growth that do not fail to implant or grow, therefore the fact that the authors of this study encounter challenges in having tumors grow in all animals injected with control cells reflect either 1) technical issues with cell viability, injection, etc. and/or 2) the fact that the off target effects of the shScr exosomes used for education are effectively inhibiting tumor growth. These issues need to be resolved before this report can be published.

The lack of tumor growth was observed in one animal, however in this animal metastatic burden was detected, indicating the B16-F10-luciferase cells were injected and aggressively spread throughout the animal. Additionally, the death observed was likely due to the aggressive growth of the B16-F10-luciferase cells. The mice that died (17 days after implantation) had tumors >1000 mm^3^ at their last measurement, which along with other surviving mice that had tumors that reached >1000 mm^3^, prompted us to stop the experiment early (18 days after implantation). As suggested by reviewer #2 we have included the shorter experimental time difference between the original study and this replication attempt in the Discussion of the revised manuscript.

Additionally, it is concerning that education of animals with 5 μg of shScr B16F10 exosomes leads to the death of some animals. This was not reported in publications with either 5 μg or 10 μg of B16F10 exosomes (for example in the original publication or in Dr. Olga Volpert's Nature Communications, 2018 report injected 10 μg of B16F10 exosomes repeatedly with no death). The deaths observed by the authors of the present study upon exosome injection are worrisome. Have the authors verified that they are measuring exosomal protein accurately after isolation? The authors should define the buffer used for exosome resuspension (in the original report was PBS as stated "The floating exosome fraction was collected again by ultracentrifugation as above, and the final pellet was resuspended in PBS"). Mouse lethality could reflect inaccurate exosome protein quantification. Alternatively, the presence of exosome aggregates in the preparation would lead to animal death, to avoid this exosomes should be vigorously resuspended before injection (10 times at least). Last but not least, the health of the cells from which the exosomes are collected can significantly influence the quality of the exosomes and could affect animal health (especially in the context of the shScr off target effects).

As mentioned above, the deaths observed were not upon exosome injection, but 17 days after B16-F10-luciferase implantation (which occurred after the 28 days of 3 times a week exosome injections). It is a possibility that exosome, or synthetic liposome, injection might have influenced this, but this does not seem likely. The more likely possibility is the aggressive growth of the B16-F10-luciferase cells, which was more severe in some animals than others.

Regarding the technical questions raised as possibly factors. Yes, we used a BCA assay, with standard curves, to quantify the amount of exosome protein after isolation using, similar to the original study. Furthermore, we characterized exosomes using Nanosight analysis. The methods defined the buffer the exosomes were resuspended in “100 µl filtered PBS (Thermo Fisher Scientific, cat# MT21040CM)”, the same buffer as the original study. While no information was included in the original paper, or during communication with the original authors regarding a specific way the exosome should be resuspended prior to injection (i.e. 10 times at least), the exosomes were resuspended right before each individual injection by pipetting 3-4 times, tapping the tube, inverted 3-4 times, and then tapping the syringe right before each injection. We have added these additional details to the Materials and methods in the revised manuscript. Also, as mentioned above, we monitored the health and qualitatively kept track of the cell growth rates of the cultures. There were also no complications during exosome injections.

Reviewer #2:Overall this replication study was carried out appropriately. I have some concerns about the quality of the data presented (in point 1) and some other comments, as follows:1) In Figure 1, The Western blots give a variety of concerns. First, they are all very dark and overblown. I would also like to see the full blots – they should probably be included in the published paper. Second, for Figure 1A, the blot comparing the shScr and shMet in the cells shows much higher Met in the shMet cells then in the controls. I presume that is a mistake? Third, is GAPDH a relevant loading control or does it change in exosomes with Met?

We have revised Figure 1, and the figure supplement, to increase the area displayed and the brightness/contrast of the images to allow for a full representation of the results. Figure 1A in the original submission was a labeling mistake that has been fixed in the revised manuscript. Gadph was the same loading control used in the original study, which is why it was used in this replication attempt. Gapdh is one of the most often identified proteins identified in exosomes (http://exocarta.org/exosome_markers). Of course, whether Gapdh is a relevant loading control, just like any other ‘housekeeping’ gene, is tricky since they vary widely, especially across sample types (e.g. Barber et al., 2005). We are not aware of any research suggesting Gapdh expression changes based on Met expression. Importantly, though we loaded equal amounts of protein (30 µg) for each sample analyzed by Western blot, opposed to adjusting amounts based on Gapdh levels.

Barber, R.D., Harmer, D.W., Coleman, R.A., Clark, B.J., 2005. GAPDH as a housekeeping gene: analysis of GAPDH mRNA expression in a panel of 72 human tissues. Physiological Genomics 21, 389–395. https://doi.org/10.1152/physiolgenomics.00025.2005

2) For Figure 1—figure supplement 1, I also don't see how a shMet can be compared to shSc when they are on different blots and it seems strange to compare the MOI to just each other and not untreated cells. This is a relatively minor concern – the assessment of Met KD in Figure 1 is the most important.

We agree that a comparison of shScr and shMet cannot be made since they are on different blots. These blots, however, illustrate that, for unknown reasons, Met levels in the shScr cells generated with an MOI of 10 were lower when compared to the shScr cells generated at the other MOI ratios.

We also agree that a comparison of shScramble and shMet cells to untransduced B16-F10 cells and exosomes would be ideal, especially considering the results observed in this replication attempt; however, this was not reported in the original study (Figure 4A of the original study is between B16-F10 and B16-F1 cells and exosomes, Supplementary Figure 5A is between B16-F10, B16-F1, and B16-F10 shMet cells and exosomes, and Figure 4E of the metastasis results is between B16-F10, B16-F10 shMet, and liposome control). It was unclear from the original paper if the B16-F10 cells and exosomes were from untransduced cells or were shScramble; however, during peer review of the Registered Report it was suggested that the B16-F10 cells reported in Figure 4E and Supplementary Figure 5A were shScramble transduced. Thus, we conducted this experiment similar to the original study design. We have revised the manuscript to highlight that the addition of untransduced B16-F10 cells would be a valuable additional control to include for future experiments to determine if any undesirable effects occurred, since infection can cause genome instability.

3) For the analysis of tumor metastasis, are the data normal or not normal? It seems like statistical analyses were used that are appropriate for data with a non-Gaussian distribution. Therefore, I'm not sure why mean and SEM are used to represent the data, which are only appropriate for data with a Gaussian distribution.

The data were normal when natural log transformed, which is what the statistical analysis was performed on. Additionally, Figure 2—figure supplement 1E, F represent the data as box and whisker plots on a natural log transformed y-axis. We agree that representing the data as mean and SEM are not appropriate (Figure 2C, D); however, with this project we presented the replication the same as the original data to allow users to directly compare results. To avoid confusion, we have reversed the figures in the revised manuscript and added a note in the figure legend. That is, the box and whisker plots are presented in the main figure and the box plots are presented in the supplement to provide a comparison to how the original data were presented.

4) For differences between this study and Peinado that are listed in the last paragraph of the Results/Discussion section of the paper, I think differences in Met signaling and in the effectiveness of knockdown (which I also doubt based on the quality of the Western blots here) should be mentioned. In addition, the study was not carried out as long as the Peinado study (18 days instead of 21 days) after exosome education, so this should be mentioned in the last paragraph too. There were also fewer mice, which could definitely affect detection of statistical significance, which should be mentioned.

We have included additional differences in the last paragraph of the revised manuscript. This replication attempt, like all of the replication attempts in the Reproducibility Project: Cancer Biology, are designed to perform independent replications with a calculated sample size to detect the originally reported effect size with at least 80% power based on the originally reported data (Errington et al., 2014). While we used fewer mice, the higher observed variance in the replication is a factor likely influencing if statistical significance is reached. This is included as a consideration when discussing the effect sizes and the RSD comparison between the original study and this replication.